# Nanopore guided annotation of transcriptome architectures

Jonathan S. Abebe,[1] Yasmine Alwie,[2] Erik Fuhrmann,[2] Jonas Leins,[2] Julia Mai,[2,3] Ruth Verstraten,[2,4] Sabrina Schreiner,[2,3,5] Angus C. Wilson,[1] Daniel P. Depledge[1,2,4,5]

**ABSTRACT** Nanopore direct RNA sequencing (DRS) enables the capture and full-length sequencing of native RNAs, without recoding or amplification bias. Resulting data sets may be interrogated to define the identity and location of chemically modified ribonucleotides, as well as the length of poly(A) tails, on individual RNA molecules. The success of these analyses is highly dependent on the provision of high-resolution transcriptome annotations in combination with workflows that minimize misalignments and other analysis artifacts. Existing software solutions for generating high-resolution transcriptome annotations are poorly suited to small gene-dense genomes of viruses due to the challenge of identifying distinct transcript isoforms where alternative splicing and overlapping RNAs are prevalent. To resolve this, we identified key characteristics of DRS data sets that inform resulting read alignments and developed the nanopore guided annotation of transcriptome architectures (NAGATA) software package (https://github.com/DepledgeLab/NAGATA). We demonstrate, using a combination of synthetic and original DRS data sets derived from adenoviruses, herpesviruses, coronaviruses, and human cells, that NAGATA outperforms existing transcriptome annotation software and yields a consistently high level of precision and recall when reconstructing both gene sparse and gene-dense transcriptomes. Finally, we apply NAGATA to generate the first high-resolution transcriptome annotation of the neglected pathogen human adenovirus type F41 (HAdV-41) for which we identify 77 distinct transcripts encoding at least 23 different proteins.

**IMPORTANCE** The transcriptome of an organism denotes the full repertoire of encoded RNAs that may be expressed. This is critical to understanding the biology of an organism and for accurate transcriptomic and epitranscriptomic-based analyses. Annotating transcriptomes remains a complex task, particularly in small gene-dense organisms such as viruses which maximize their coding capacity through overlapping RNAs. To resolve this, we have developed a new software nanopore guided annotation of transcriptome architectures (NAGATA) which utilizes nanopore direct RNA sequencing (DRS) datasets to rapidly produce high-resolution transcriptome annotations for diverse viruses and other organisms.

**KEYWORDS** nanopore, direct RNA sequencing, transcriptome, annotation, adenovirus, herpesvirus, coronavirus, HAdV-F41

T he transcriptome architecture of a given organism denotes the full catalog of RNAs arising from the combined action of transcription and post-transcriptional processing. Of these, many RNAs are transcribed only in specific temporal or tissue contexts or in response to intrinsic or extrinsic stresses. The content and complexity of transcriptome architectures vary dramatically between different organisms and can be broadly classified as gene sparse or gene dense depending on the proportion of the genome that encodes transcripts. In contrast to the large gene-sparse genomes of most eukaryotes and archaea, the genomes of viruses are generally small and gene-dense

Address correspondence to Daniel P. Depledge, depledge.daniel@mh-hannover.de.

Yasmine Alwie and Erik Fuhrmann contributed equally to this article.

The authors declare no conflict of interest.

See the funding table on p. 17.

(1). This poses a significant challenge to studies of gene regulation, transcription, and translation, particularly when using short-read sequencing approaches as these cannot adequately resolve alternative splicing and overlapping RNAs (2).

Long-read RNA sequencing enables the sequencing of full-length RNAs in the form of both native and recoded (cDNA) RNA using platforms developed by Oxford Nanopore Technologies and Pacific Biosciences (3). These methodologies have significantly enhanced our ability to annotate transcriptomes of all sizes and complexities by (i) resolving simple and complex repeat regions, (ii) providing linkage between splice sites in studies of alternative splicing, and (iii) enabling the discovery of new transcript isoforms. The specific attraction of nanopore direct RNA sequencing (DRS) (4), is the power to interrogate RNA biology at the level of individual molecules. In theory, each sequence read derived by DRS represents a single native RNA and thus contains all the information needed to identify (i) the corresponding genomic sequence from which it was transcribed, (ii) all modified ribonucleotides within the RNA molecule, and (iii) the length of the poly(A) tail (if present). This information can in turn guide predictions of secondary structure, stability, and ultimately, function. Our ability to perform such comprehensive analyses is steadily increasing with the development of computational approaches to extract such data (5–10). However, to successfully interrogate RNAs at the level of individual molecules, it is crucial that sequence reads can be unambiguously assigned to the correct transcript isoform—a process that requires a high-resolution annotation of the underlying transcriptome architecture. This has been demonstrated in a number of recent studies, all of which required the generation of high-resolution transcriptome annotations to facilitate the desired analysis (11–16). While many of these high-resolution annotations were obtained by laborious manual processing, this is neither a practical nor sustainable methodology. Several computational approaches capable of providing high-resolution transcriptome annotations and quantifications have recently been developed and have proven extremely powerful in the context of studying the gene sparse transcriptomes of higher eukaryotes (17). Examples include Stringtie2 (18), Bambu (19), and Isoquant (20). However, as these approaches appear designed with higher eukaryotic transcriptomes in mind, their utility in decoding the gene-dense transcriptomes of viruses remains poor. This remains a significant issue for many viral pathogens including the adenovirus strain F serotype 41 (HAdV-F41) which is the primary cause of adeno-associated acute gastroenteritis of infants (21–23) and more recently has been associated with adeno-associated virus-driven cases of acute liver failure (24). HAdV-F41 differs from other human adenoviruses in terms of tropism and a detailed examination of its transcriptome and protein-coding potential is urgently needed to provide further insight into its molecular behavior and pathogenicity.

To resolve this, we have developed a new computational approach entitled nanopore guided annotation of transcriptome architectures (NAGATA) and showcase its ability to generate high-resolution transcriptome annotations from DRS data sets. Using both synthetic and real nanopore data sets, we demonstrate that NAGATA significantly outperforms other annotation tools in accurately reconstructing the transcriptomes of selected DNA and RNA viruses. We further present a new high-resolution transcriptome annotation for the neglected human pathogen, HAdV-F41.

## MATERIALS AND METHODS

### Publicly available data sets used in this study

Raw fast5 data sets for HAdV-C5 (PRJEB35667), Varicella Zoster Virus (VZV) (PRJEB38829), and hCoV-OC43 (PRJEB42052) were downloaded from the Sequence Read Archive (https://www.ncbi.nlm.nih.gov/sra).

### Reference genomes and source annotations

The human genome assembly (GRCh38.p14) and GTF annotation files were obtained from Ensembl (https://www.ensembl.org/index.html). All viral reference genomes were

downloaded from Genbank (https://www.ncbi.nlm.nih.gov/genbank/). The following accession numbers were used: HAdV-C5 (AC_000008.1), HAdV-F41 (ON561778.1), VZV strain Dumas (NC_001348.1), and hCoV-OC43 (NC_006213.1). The corresponding GFF3 annotation for HAdV-F41 was downloaded from the same source. GFF3 annotations for HAdV-C5, SVV, and hCoV-OC43 were obtained from repositories associated with the recent reannotation efforts (12, 14, 25).

## Generation of *in silico* data sets

Strand-separated GFF3 files were converted, via genePred files, to BED12 files using UCSC tools (26). BEDtools v2.27.1 (27) getfasta [ *-s -split* ] was used to generate multi-FASTA files containing 200 copies of each transcript.

## Generation of HAdV-F41 nanopore DRS data sets

A549 cells (ATCC, No. CCL‑185) and HEK293 (ECACC European Collection of Authenticated Cell Cultures; Sigma-Aldrich, No. 85120602-1VL) were grown in Dulbecco's modified Eagle's medium supplemented with 5% fetal calf serum, 100 U of penicillin, 100 µg of streptomycin per mL in a 5% CO2 atmosphere at 37℃. These cell lines are frequently tested for mycoplasma contamination. HAdV-F41 [wild-type Tak strain (28)] was propagated and titrated in HEK293 cells by quantitative immunofluorescence staining of the hexon protein (8C4, Santa Cruz Biotechnology) at 48 hours post-infection (hpi) as previously published (29). For HAdV-F41 infection, A549 cells were infected at a multiplicity of infection (MOI) of 50 in non-supplemented Dulbecco's Modified Eagle Medium (DMEM). After incubation for 1 h at 37℃, infection was stopped by replacing the virus-containing medium with fresh DMEM medium with supplements. At defined times post-infections, the supernatant was removed and cells were lysed in 8 mL Trizol per 10 cm dish. After equilibration, 0.2 vol of chloroform was added followed by vigorous vortexing, a 3-min incubation at RT, and centrifugation at 12,000 × *g* for 15 min at 4℃. The aqueous phase was collected and precipitated using 0.5 vol isopropanol and 1 uL of Glycoblue (Invitrogen) for 10 min at RT prior to pelleting by centrifugation at 12,000 × *g* for 15 min at 4℃. Pellets were washed in 75% ethanol and centrifuged at 12,000 × *g* for 5 min at 4℃. The supernatant was removed and the pellet air-dried for 5 min before resuspending in RNAse-free water and incubated at 55℃ for 10 min before quantification with a Qubit hsRNA kit (Invitrogen). Poly(A) selection was performed using Dynabeads (Invitrogen) with 133 µL beads added to 25 µg of total RNA. Nanopore DRS libraries were prepared according to the Deeplexicon multiplexing protocol (30) and sequenced for 24 h on an R 9.4.1 flowcell using a MinION Mk.1b.

## Basecalling and poly(A) tail estimation

For all DRS data sets, high-accuracy basecalling was performed with Guppy v6.5.7 [ *-c rna_r9.4.1_70bps_hac.cfg -r --calib_detect --trim_strategy rna --reverse_sequence true* ] and poly(A) tail analyses generated with nanopolish v0.14.

## Alignment and downstream processing of *in silico* and DRS data sets

For HAdV-C5, VZV, and HG38, reads in fastq files were aligned against the respective reference genome using minimap2 (31) [ *-ax splice -k14 -uf --secondary=no* ] and parsed to generate sorted BAM files using SAMtools v1.15 (32) in which only primary alignments were retained [ *samtools view -F 2308* ]. For hCoV-OC43, alternative minimap2 parameters were specified [ *-ax splice -k 8 -w 3 -g 30000 -G 30000 -C0 -uf –no-end-flt –splice-flank=no* ] to account for discontinuous transcription (33).

## Transcriptome reconstruction parameters

To reconstruct the transcriptomes presented in this study, NAGATA was run with the following (default) parameters [ *-s 5 -c 100 -t 50 -cg 50 -tg 30 -iso 15 -m 8 -a*

*1 -b 1* ], except for several instances in which -c and -t parameters were reduced. For Stringtie2 v2.1.3 (18), the following flags were used [ *--viral -L* ]. For Isoquant v3.1.1 (20) [ *--data_type nanopore --model_construction_strategy default_ont --splice_correction_strategy default_ont, --fl_data --matching_strategy loose --report_novel_unspliced* ]. For Bambu v3.3.5 (19), we followed the *de novo* transcript discovery approach described in their manual and included a flag for single exon discovery [ *bambu(reads = "in.bam," annotations = NULL, genome = "ref.fasta," NDR = 1, quant = FALSE, opt.discovery = list(min.txScore.singleExon = 0))* ]. For all four tools, the same sorted BAM files were used as input, and resulting GTF files (Stringtie2, Isoquant, Bambu) were converted into bed files using UCSCutils (26) gtfToGenePred and genePredtoBed. Resulting BED and BAM files were visualized using the integrative genomics viewer (IGV)(34)

## Overlap analyses

Transcript annotations produced by each tool were converted from GFF3/GTF to BED12 using gtfToGenePred and genePredToBed from the UCSCutils (26) package and compared against existing annotations in BED12 format using the custom Python script post_intersect_processing_v4.1.py. This produces three BED12 outputs: annotation overlaps, tool-specific annotations, and annotation only. Each of these contains transcripts assigned to these three categories. To compare outputs from multiple tools (e.g., annotation overlaps from NAGATA, Bambu, Isoquant, and Stringtie2), we used the custom Python script multiple-overlap.v1.py. Both custom scripts are available from https://github.com/DepledgeLab/NAGATA. F1 scores [$2 \times (P \times r)/(P + r)$] were calculated using (*P*) precision [true positives (TP) + false positives (FP) + false negatives (FN)] and (*r*) recall [TP/(TP + FN)] values.

## Generation of R plots and R packages used

All plotting was performed using Rstudio (https://posit.co/download/rstudio-desktop/) with R v4.1.1 and the following packages: data.table (https://r-datatable.com), Gviz (35), GenomicFeatures (36), ggplot2 (37), UpSetR (38), dplyr (https://dplyr.tidyverse.org/), tidyr (https://tidyr.tidyverse.org/), and patchwork (https://github.com/thomasp85/patchwork).

## RESULTS

### Characteristics of nanopore DRS genome alignments inform transcript boundaries

The aim of this study was to implement a new algorithm for generating high-resolution transcriptome annotations from DRS data sets using read alignments against a genome of interest. As standard DRS proceeds in a 3′ → 5′ direction, first through the adapter, then the poly(A) tail, and finally the body of the RNA itself, all reads are expected to contain the poly(A) tail and the 3′ end of the RNA. Processing of the raw nanopore signals allows the segmentation of these three units (4). This, in theory, allows precise plotting of the cleavage and polyadenylation sites (CPAS). However, an analysis of multiple extant nanopore DRS data sets (14, 39, 40) using nanopolish (5) demonstrates that poly(A) tails can only be reliably detected in ~58%–83% of the reads (Table S1). We theorized that reads for which a poly(A) tail could not be identified by nanopolish would likely be over- or under-trimmed and that this would impact on accurate mapping of CPAS. For the 5′ end of RNAs, it has been previously reported that nanopore DRS cannot sequence the ~5–10 terminal nucleotides due to the presence of the m7G cap (14, 15, 41–43). This feature is irrespective of the underlying RNA source and results in estimated rather than precise transcription start sites (TSS). Given the continuous turnover of poly(A) RNA in the cell, combined with *in vivo/vitro* strand breakage and signal processing errors (41), only a proportion of sequenced RNAs are expected to be full length and thus would share near-identical 5′ alignment ends that can be interpreted as TSS. For non-full-length RNAs that originate from multi-exon splicing, this can create alignment artifacts where 5′ ends cannot be extended across splice junctions. This in turn leads to extensive 5′ soft

clipping of the alignment and the clustering of many 5′ alignments ends at the same location, thus giving rise to artifact TSS.

To examine this more closely, we used data sets that we previously generated from adenovirus type 5 (HAdV-C5) infected A549 cells and VZV-infected ARPE-19 cells for which high-resolution transcript annotations exist and for which the TSS and CPAS have been confirmed by orthologous methodologies (12, 14). We segregated reads according to the presence or absence of a detectable poly(A) tail and whether resulting alignments showed 5′ soft clipping >3 nt, and subsequently determined the closest annotated TSS and CPAS for each read. Soft clipping denotes portions of a read that cannot be aligned to the target, either due to sequence mismatch or, in the case of splice junctions, the inability to locate the 5′ junction site. For both HAdV-C5 and VZV we observed that reads with 5′ soft clipping >3 nt could be associated with artifact TSS and produced high levels of noise in regions proximal to previously confirmed TSS (Fig. 1A and B; Fig. S1A and B). Similarly, alignments using reads without detectable poly(A) tails resulted in larger numbers of 3′ alignments that were >50 nt from defined CPAS (Fig. 1C; Fig. S1C). Note that the CPAS used for HAdV-C5 and VZV were previously defined using Illumina RNA-Seq data sets (12, 14) in conjunction with ContextMap2 (44). TSS used for HAdV-C5 were previously defined by nanopore DRS while those used for VZV were defined by CAGE-Seq (12, 14) The latter is considered the most accurate method as nanopore DRS can only sequence to within 5–10 nt of the 5′ cap (14, 15), hence why the "distance to nearest TSS" shows a greater offset in the VZV data (Fig. S1A and B). Thus, defining TSS and CPAS using DRS alignments requires careful filtering of reads without measurable poly(A) tails and alignments showing 5′ soft clipping, a procedure that is not currently utilized by existing transcriptome annotation software.

## Nanopore guided annotation of transcriptome architectures

Utilizing the characteristics described above, the NAGATA algorithm is designed to convert DRS alignments against a genome into corresponding transcriptome annotations. As input, it accepts a sorted BAM file containing genome-level primary alignments and a poly(A) output file from the nanopolish package (5). NAGATA subsequently functions through three distinct stages (i) pre-filtering, (ii) TSS/CPAS definition, and (iii) isoform deconvolution and filtering (Fig. 2). The pre-filtering step masks alignments for which (i) a poly(A) tail could not be detected by nanopolish, and/or (ii) soft clipping above a specified threshold (default = 3) is observed at the 5′ ends of the alignment. Using the top strand of the recently reannotated adenovirus type 5 (HAdV-C5) transcriptome as an example (12), we demonstrate how raw DRS alignments lead to multiple artifacts of TSS and CPAS that are otherwise eliminated when applying our pre-filtering strategy (Fig. 2A and B). For the TSS/CPAS definition step (Fig. 2C), NAGATA first defines transcriptional units (TUs) by grouping alignments with identical 3′ ends and determining the number of alignments in each group. Alignments generated from reads without detectable poly(A) tails are masked at this stage while reads with 5′ soft clipping are retained. 3′ end positions with an abundance count above a user-defined threshold (default = 50) are used as anchors and all alignments with 3′ ends within a defined distance (default ± 25 nt) of an anchor are added to the group. If two or more anchors are present in this range, the defined 3′ end defaults to the anchor with the largest number of initial alignments (Fig. 3). Once 3′ end grouping is complete, each anchor is defined as a CPAS and the 3′ end of individual alignments in each group are corrected to match that of the anchor (Fig. 3). This process is subsequently repeated to define TSS by grouping the 5′ ends of all alignments. 5′ alignments within a defined distance (default ± 12 nt) of an anchor are added to the same group and the 5′ ends of individual alignments are corrected to match that of the anchor (Fig. 3). Here, alignments generated from reads with 5′ soft clipping are masked while reads without detectable poly(A) tails are retained. The final step deconvolutes the isoforms present in each TU (Fig. 2D), a function that is performed by first segregating alignments by the number of exons present and subsequently by comparing the genomic position of the exons. Here a distance of up to 50 nt

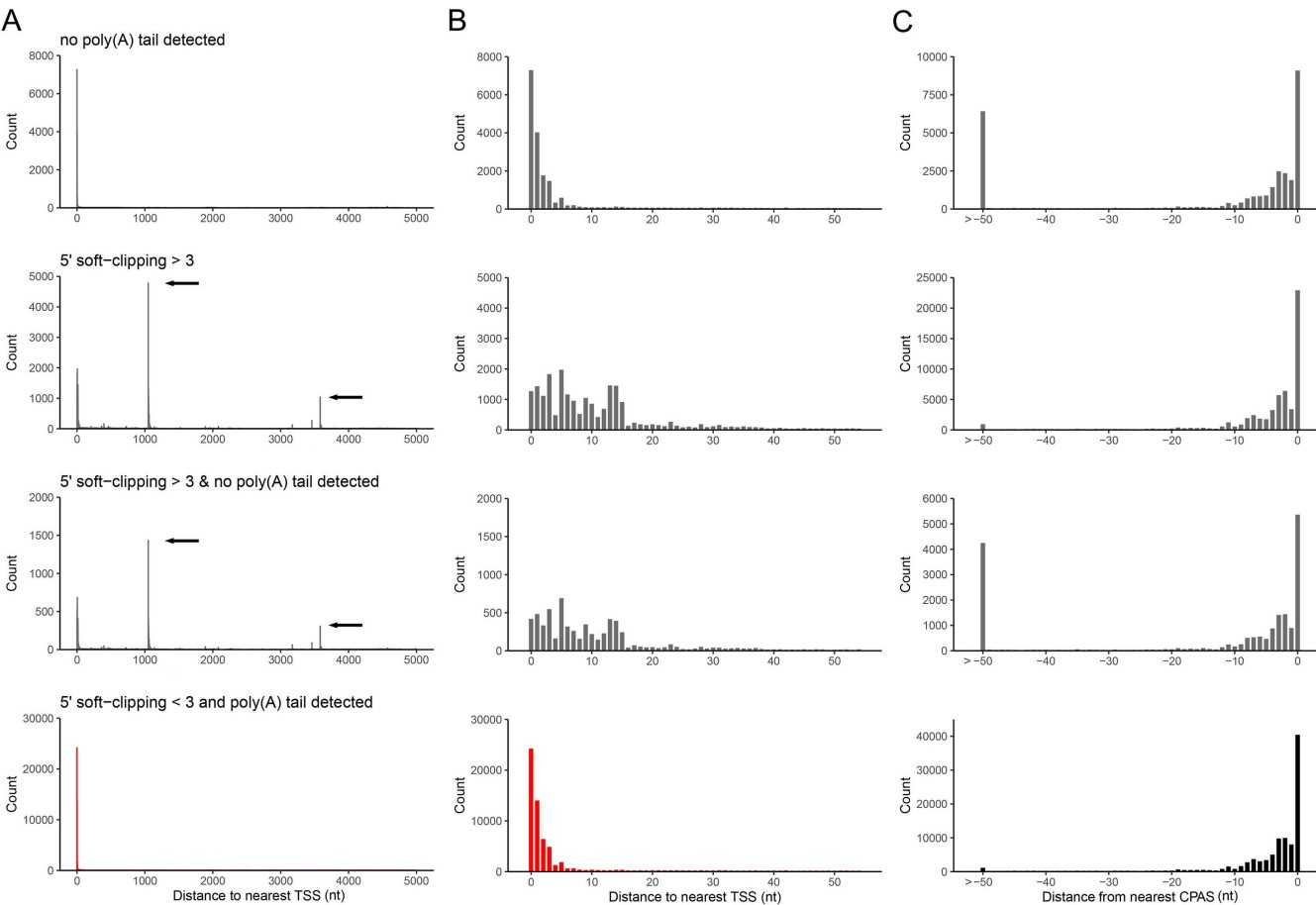

**FIG 1** Characteristics of nanopore DRS alignments. Alignments of adenovirus type 5 DRS reads were segregated according to the presence/absence of detectable poly(A) tails and the presence of soft-clipping values >3 at the 5′ end. (A–C) The genomic location and read count of (A and B) 5′ alignment ends relative to previously defined TSS or (C) 3′ alignment ends relative to previously defined CPAS were determined for each of four conditions (no poly(A) tail detected, 5′ soft-clipping >3, no poly(A) tail and 5′ soft-clipping >3, and poly(A) tail and 5′ soft-clipping ≤3) across windows of (A) 5,000 nt and (B and C) 50 nt. Black arrows indicate the location of artifact TSS derived from misalignment across splice junctions.

between exon start and end positions is allowed between alignments, again with a correction step based on the position with the most alignments. Finally, for each resulting isoform, we apply two filters to decide on the validity of the isoform. The first filters on the total number of supporting alignments (raw count) while the second calculates a TSS/CPAS ratio (number of supporting alignments/total alignments associated with the same TSS/CPAS). The latter specifically functions to identify and remove low abundance isoforms (by default <1% frequency).

## Benchmarking NAGATA using synthetic data sets

Transcriptomes vary in size and complexity between organisms but most analytical softwares appear designed and optimized for a specific organism, e.g., *Homo sapiens*, a process that may lead to suboptimal performances when applied to different transcriptome architectures. The aim of NAGATA was to implement an approach that is agnostic in regard to the underlying transcriptome architecture. To test this, we generated an *in silico* data set comprising 200 copies of all RNAs encoded on chromosome 1 of the gene sparse human genome and aligned these back against the genome using Minimap2 (31). We calculated precision and recall (F1) scores for NAGATA and three popular annotation tools; Stringtie2 (18), Isoquant (20), and Bambu (19), and observed all four produced similar results (Fig. 4A). Surprisingly, no tool achieved a perfect score indicating that even

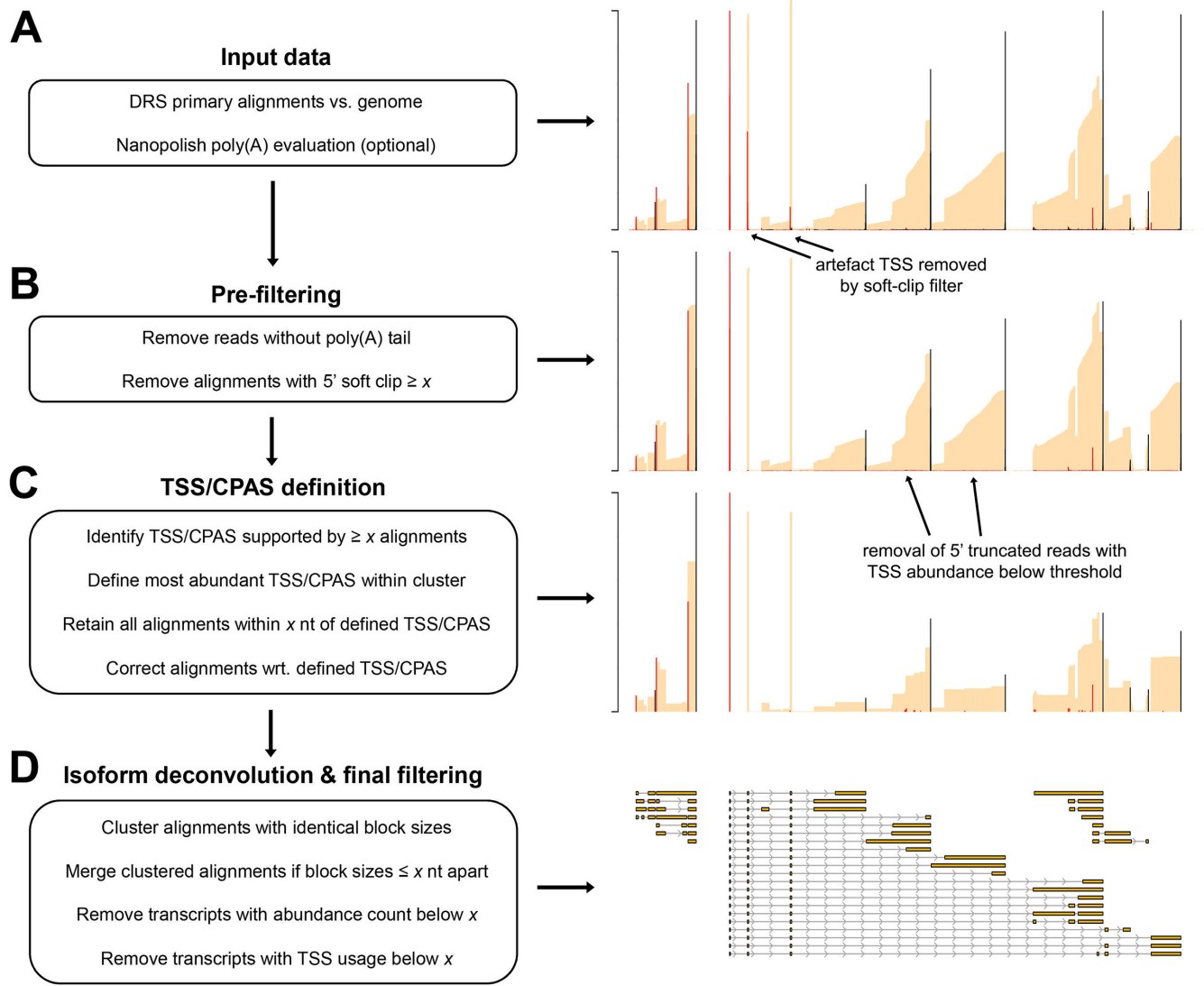

**FIG 2** Overview of the NAGATA methodology. (A) DRS genome alignments (beige) that are filtered to retain only primary mappings and (optionally) nanopolish poly(A) output files are used as input for NAGATA. Putative TSS (red) and CPAS (black) are defined (TSS/CPAS definition) by counting the number of alignments with identical 5′ (TSS) or 3′ (CPAS) ends. (B) Pre-filtering removes alignments with 5′ soft-clipping values greater than a specified value and optionally removes read alignments for which poly(A) tails are not detected by nanopolish. (C) TSS and CPAS are defined (TSS/CPAS definition) by counting the number of alignments with identical 5′ (TSS) or 3′ (CPAS) ends and considering only those exceeding a specified count as valid. For TSS/CPAS that pass this threshold, all neighboring TSS/CPAS within a defined distance are retained and their coordinates adjusted to the dominant TSS/CPAS position. At this stage, TU is defined and all alignments sharing the same CPAS are considered part of the same TU (i.e., transcripts with differing TSS but the same CPAS are considered part of the same TU). (D) For each resulting TU, transcript isoform deconvolution and final filtering are performed by first collapsing alignments if they share the same blockSize and blockStarts distribution and only those exceeding a specified count are considered valid. Alignments with similar blockSize/blockStart values (typically within 1–3 nt) are merged prior to filtering based on abundance counts. Finally, NAGATA applies a filter to remove transcripts with a TSS usage below a defined fraction.

with an idealized data set, the process of alignment alone introduces artifacts and error into the final results. We next applied the same strategy to two DNA viruses (Adenovirus Type 5 and VZV) with distinct gene-dense transcriptome architectures (12, 14). HAdV-C5 transcriptomes consist of relatively few TUs and large numbers of alternatively spliced RNAs (Fig. S2), whereas VZV transcriptomes consist of large numbers of TUs, each predominantly comprised of multiple single-exon transcripts with unique TSS but shared 3′ co-terminal ends (Fig. S3). NAGATA produced an F1 score of 0.99 for HAdV-C5 and was able to reconstruct 88/89 transcript isoforms with no false positives (Fig. 4B; Fig. S2). Both

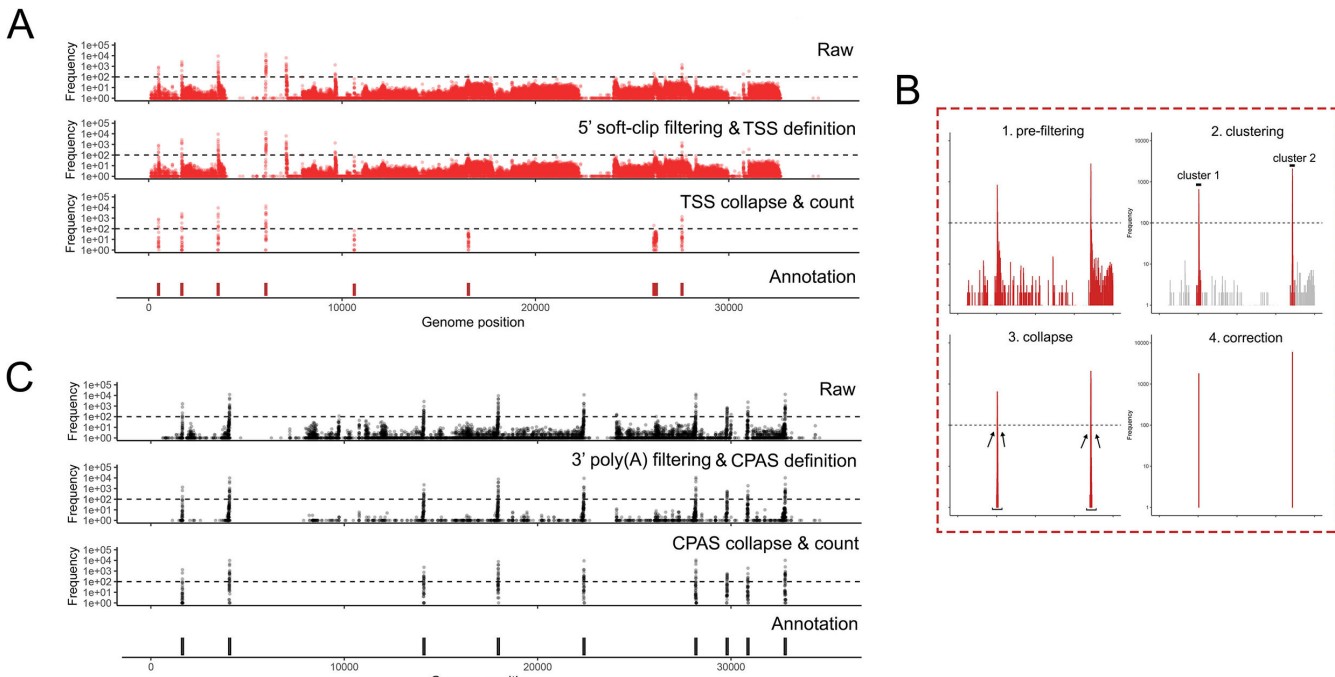

**FIG 3** Pre-filtering at 5′ and 3′ ends defines robust TSS and CPAS. NAGATA masks alignments with 5′ soft-clipping values above a user-defined value (default = 3) and, where nanopolish poly(A) output files are available, also removes alignment derived from reads without detectable poly(A) tails. (A) Visualization of TSS positions contained in the raw alignment (top), pre-filtered alignments, i.e., post-5′ soft-clipping and 3′ poly(A) tail filtering (second row), post-collapse of neighboring TSS (third row), and the existing TSS/CPAS annotation position (bottom row). (B) A specific close-up showing the pre-filtering, clustering, collapse, and correction steps. (C) Same as (A) but for CPAS.

Stringtie2 (77/89 true positives, 13 false positives, F1 = 0.83) and Isoquant (71/89 true positives, 19 false positives, F1 = 0.79) were able to identify the relative positions of all canonical TSS and CPAS, apart from Stringtie2 failing to correctly identify transcripts of the E3 region (Fig. S2). Instead, the predicted transcripts have similar structures but were coordinated and shifted relative to the canonical transcripts. Bambu performed the least well (31/89 true positives, 0 false positives, F1 = 0.34). While Bambu and Isoquant were both run using parameters allowing for the detection of mono-exonic transcripts (e.g., pIX, E3.12k, and E4orf1), neither tool was able to identify any. For VZV, NAGATA produced an F1 score of 0.97 with 135/137 transcript isoforms correctly identified and three false positives (Fig. 4C; Fig. S3). Stringtie2 (55/137 true positives, 3 false positives, F1 = 0.40), Isoquant (6/137, 0 false positives, F1 = 0.04), and Bambu (23/137, 25 false positives, F1 = 0.15) all performed poorly. Together, these results indicate that the underlying architecture of the selected gene-dense viral transcriptomes can be resolved by NAGATA but not by other existing annotation tools.

## Benchmarking NAGATA using real nanopore data sets

To verify that the results shown above were not biased by using synthetic (idealized) data sets, we next examined NAGATA's performance using real DRS data sets. We first downloaded a subset of the DRS data used to generate the most recent annotation of HAdV-C5 (12). These data sets were derived from A549 cells infected with HAdV-C5 for either 12 or 24 h and were analyzed individually (12 h, 24 h) and in combination (12–24 h). Using the 12–24 h combined data set, NAGATA identified 144 transcripts, 71 of which were present in the existing annotation (Fig. 5A). Of the 73 novel transcripts identified by NAGATA, the majority could be classified as either incompletely spliced pre-mRNAs or spliced isoforms of known transcripts. Taking the E1 region as an example, NAGATA identified 11/13 annotated transcripts and two "novel" transcripts that we classified as

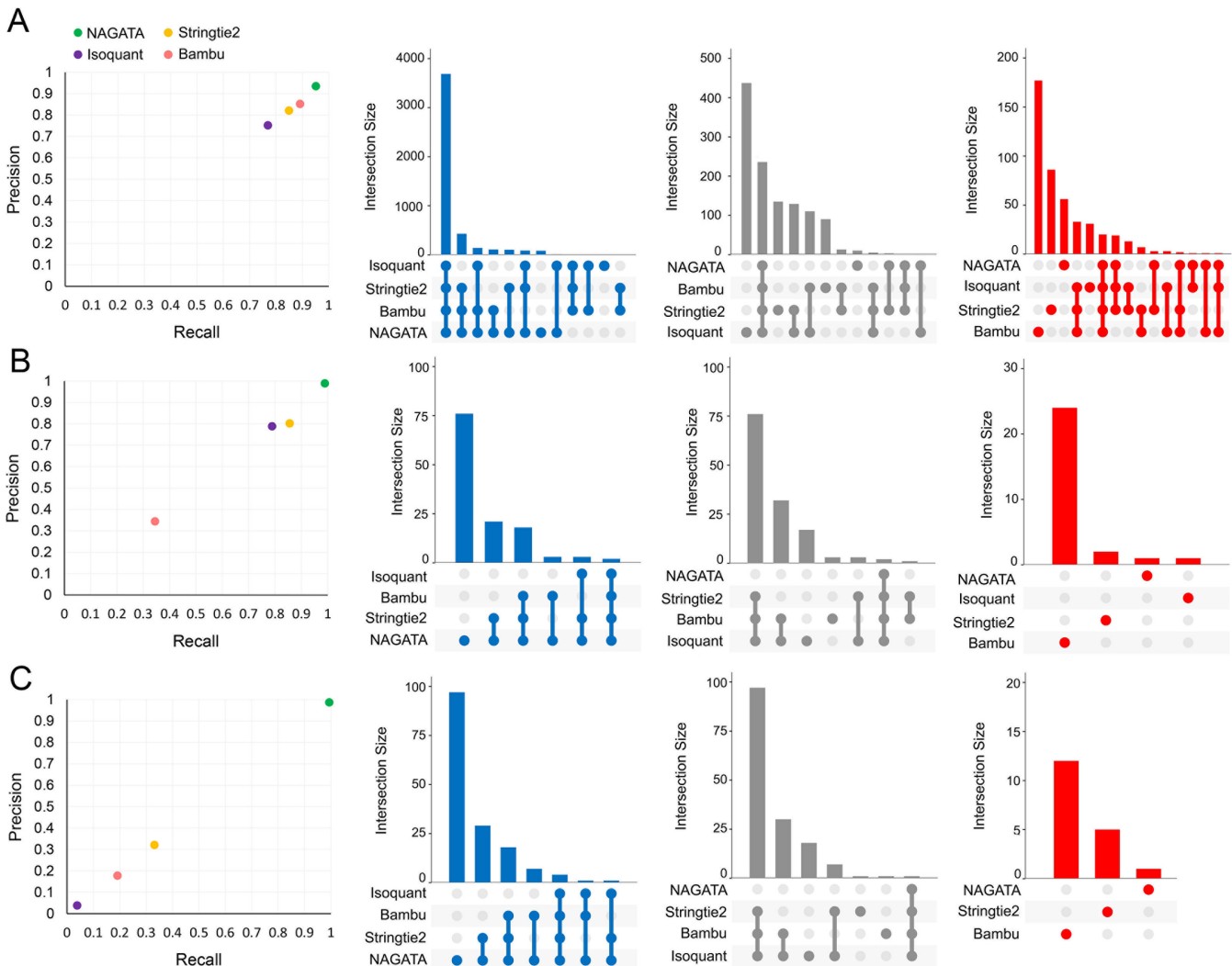

**FIG 4** Benchmarking NAGATA using synthetic data sets. Scatter plots comparing precision and recall values for NAGATA and three additional softwares [StringTie2 (18), Isoquant (20), and Bambu (19)] capable of *de novo* transcriptome annotation are supported by UpSet plots denoting the breakdown of true-positives (blue), false-negatives (gray), and false-positives (red) are shown for (A) *H. sapiens* HG38 assembly chromosome 1, (B) Adenovirus type 5 (HAdV-C5), and (C) VZV data sets.

unspliced polyadenylated pre-mRNAs of comparatively low abundance (Fig. 5A and B). In total 13/73 transcripts were recorded as incompletely spliced pre-mRNAs (marked with asterisks in Fig. 5A, the majority located in the L1 region). A further 16/73 novel transcripts matched the existing transcript structure but additionally contained the "i-leader" exon that is occasionally incorporated into the Major Late Promoter tripartite leader (45). The remaining 44 newly identified transcripts were alternatively spliced isoforms of previously annotated transcripts. Notably, all were of relatively low abundance within a given transcription unit (Fig. 5B). A further 16 transcripts in the current annotation were not detected. Visual inspection of the raw read data confirmed them to be either absent in that specific data set or supported by only 1–2 reads and thus below the detection threshold of NAGATA. To examine the value of including multiple time points when running NAGATA, we compared the results obtained from the individual 12 h (*n* = 75 transcripts) and 24 h (*n* = 125 transcripts) data sets with the 12–24h data set (*n* = 144 transcripts) (Fig. 5C). Unsurprisingly, merging of the data sets increased the number of transcripts reported. To measure the impact of overall sequencing depth on NAGATA results, we randomly subsampled the HAdV-C5-12h-24h data set to four different read

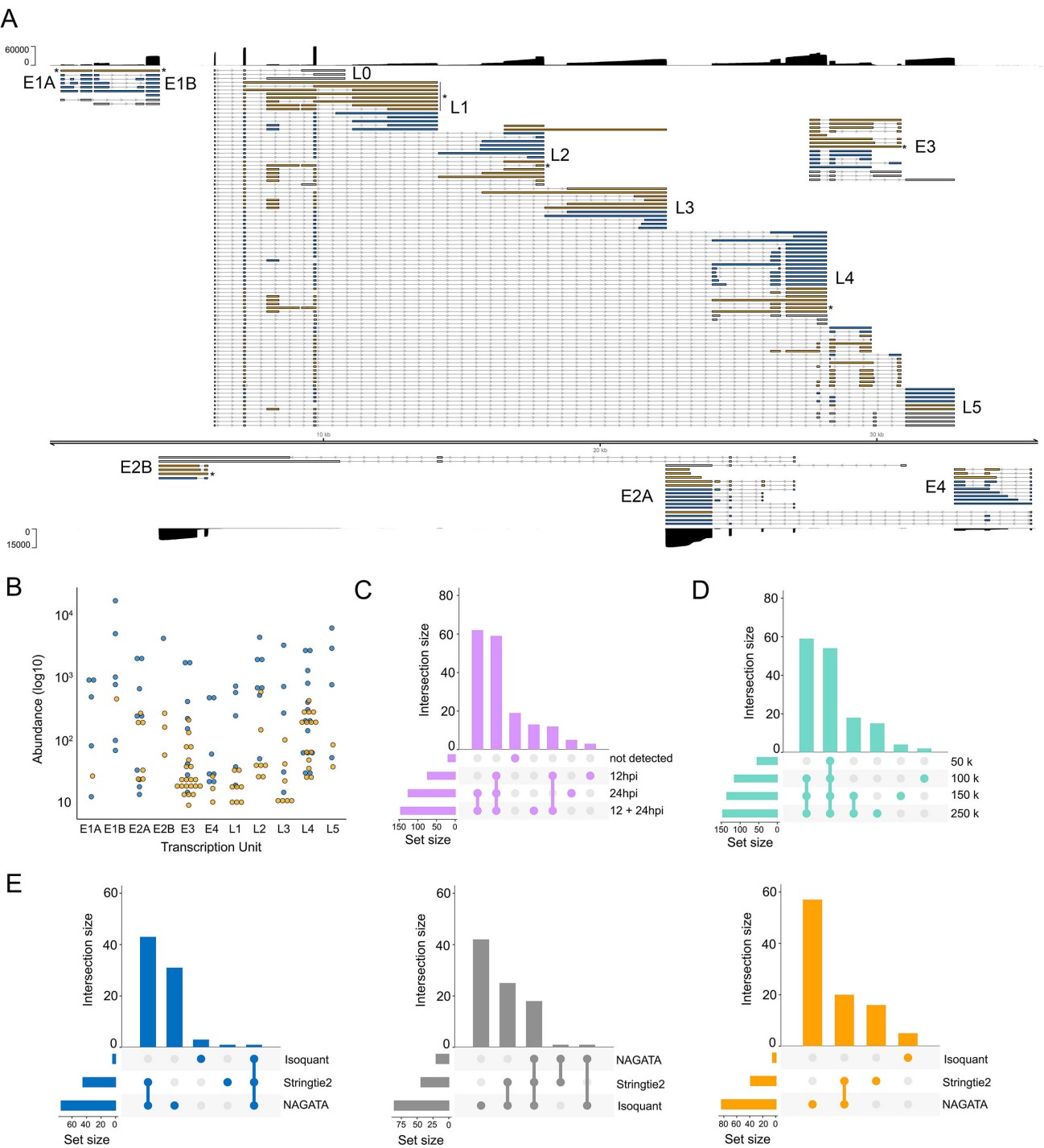

**FIG 5** Reconstruction of the adenovirus type 5 transcriptome. (A) Schematic depicting the HAdV-C5 transcriptome as constructed by NAGATA using merged DRS data sets representing 12 and 24 hpi of A549 cells. Read coverage is shown for each strand (black) with the y-axis denoting read depth. Major transcription units (e.g., E1A) are indicated in black text while individual transcripts are colored according to classification (orange = not present in existing annotation, blue = present in existing annotation, gray = reported in existing annotation but not detected by NAGATA in this data set). Wide and thin boxes indicate canonical CDS domains and UTRs, respectively. Black asterisks denote transcripts putatively classified as unspliced pre-mRNAs. (B) For each detected transcript in each transcription unit, a raw abundance count was generated using NAGATA and color-coded according to transcript classification. (C) Upset plot denoting the number of transcripts reported by NAGATA in the individual 12 hpi and 24 hpi data sets and merged 12 + 24 hpi data sets. (D) Upset plot denoting the number of transcripts reported by NAGATA in the merged 12 + 24 hpi data set after random subsampling of reads. (E) Upset plots denoting the number of transcripts reported by NAGATA, StringTie2 (18), and Isoquant (20), segregated according to (blue) overlaps with existing annotation, (gray) not detected, and (orange) not present in original annotation.

depths and applied NAGATA using the default parameters. Intriguingly, the number of transcripts identified showed only a small increase between 100 k to 250 k viral reads, suggesting that subsampling approaches may be useful for identifying when sequencing depth has reached a saturation point for transcript detection (Fig. 5D). Finally, we again compared NAGATA's performance to that of Stringtie2 and Isoquant and observed a large reduction in the numbers of transcripts identified that overlapped with the existing annotation and, in the case of Stringtie2, a number of novel transcripts that did not overlap with the novel transcripts identified by NAGATA (Fig. 5E; Fig. S4) and were not supported by the underlying read alignments.

For the second test, we downloaded and analyzed a VZV DRS data set that was derived from ARPE-19 cells infected at low MOI with wild-type VZV strain EMC-1 for 96 h (13). We aligned this data set against the VZV strain Dumas genome and processed the output with NAGATA, comparing the results against the existing VZV transcriptome annotation (13) (Fig. 6A). Following visual inspection of the pre-filtered data sets (i.e., after removal of reads without well-defined poly(A) tails and removal of alignments with 5′ soft-clipping values >3), we reduced the thresholds for defining putative TSS (-t flag) and CPAS (-c flag) which increased the number of putative TSS peaks from 67 to 89 and CPAS peaks from 44 to 54 (Fig. 6B; Fig. S5). This approach increased the total number of transcripts reported by NAGATA from 129 using default settings to 147 using optimized settings (Fig. 6C). A corresponding increase in the number of transcripts overlapping with existing annotations (from 58 to 76) was also observed (Fig. 6C). We examined the read alignments underlying the new transcripts ($n = 71$, Fig. 6A) and confirmed 24 of these utilized 22 distinct TSS that were not previously reported while just two utilized CPAS that had not previously been described. The remaining new transcripts could be classified as alternatively spliced or single exon transcripts that utilized different combinations of existing TSS and CPAS. Visual inspection of the read data confirmed the newly identified TSS to be robust and further confirmed an absence of sufficient read data at previously reported TSS that were not identified by NAGATA. A total of 56 transcripts from the original annotation were not identified here. Notably, many of these were located in regions of low read coverage and thus were either not supported by enough individual reads or were absent entirely in the downloaded data set. Of transcripts encoding known protein-coding ORFs, only four were not detected by NAGATA in this data (pORF28, pORF38, pORF55, and pORF56) (Fig. 6A). Across all reported transcripts, the abundance value of newly identified transcripts (median read count = 90) was lower than for previously annotated transcripts (median read count = 300) (Fig. 6D). Further analyses with Stringtie2 and Isoquant again resulted in a small number of overlapping transcripts being identified and a large number of erroneous transcripts that were not consistent with the underlying read alignments (Fig. 6E; Fig. S6).

## Application of NAGATA to a cytoplasmic RNA virus

To expand beyond nuclear-replicating DNA viruses, we also examined the ability of NAGATA to reconstruct the transcriptome of the cytoplasmic betacoronavirus hCoV-OC43. Coronaviruses are members of the Nidovirales order which replicate through transcription of negative-sense RNA intermediates that serve as templates for positive-sense genomic RNA (gRNA) and sub-genomic RNAs (sgRNAs). sgRNAs are generated through a process termed discontinuous transcription that combines a leader sequence in the 5′ UTR with varying regions from the 3′ end of the genome (46). From a computational perspective, alignments of sgRNAs against a genome appear similar to those generated from spliced RNAs although care must be taken to ensure that alignment and downstream processing software accurately record the junctions between the leader sequence and body. The prior annotation of hCoV-OC43 identified nine sgRNAs in addition to the primary gRNA (47). Using a publicly available DRS data set that was generated from hCoV-OC43 infected MRC-5 cells (25), we observed that NAGATA was able to reconstruct all reported sgRNAs in addition to two previously unannotated sgRNAs (Fig. 7A). Of these, both of which were low abundance (Fig. 7B), the first

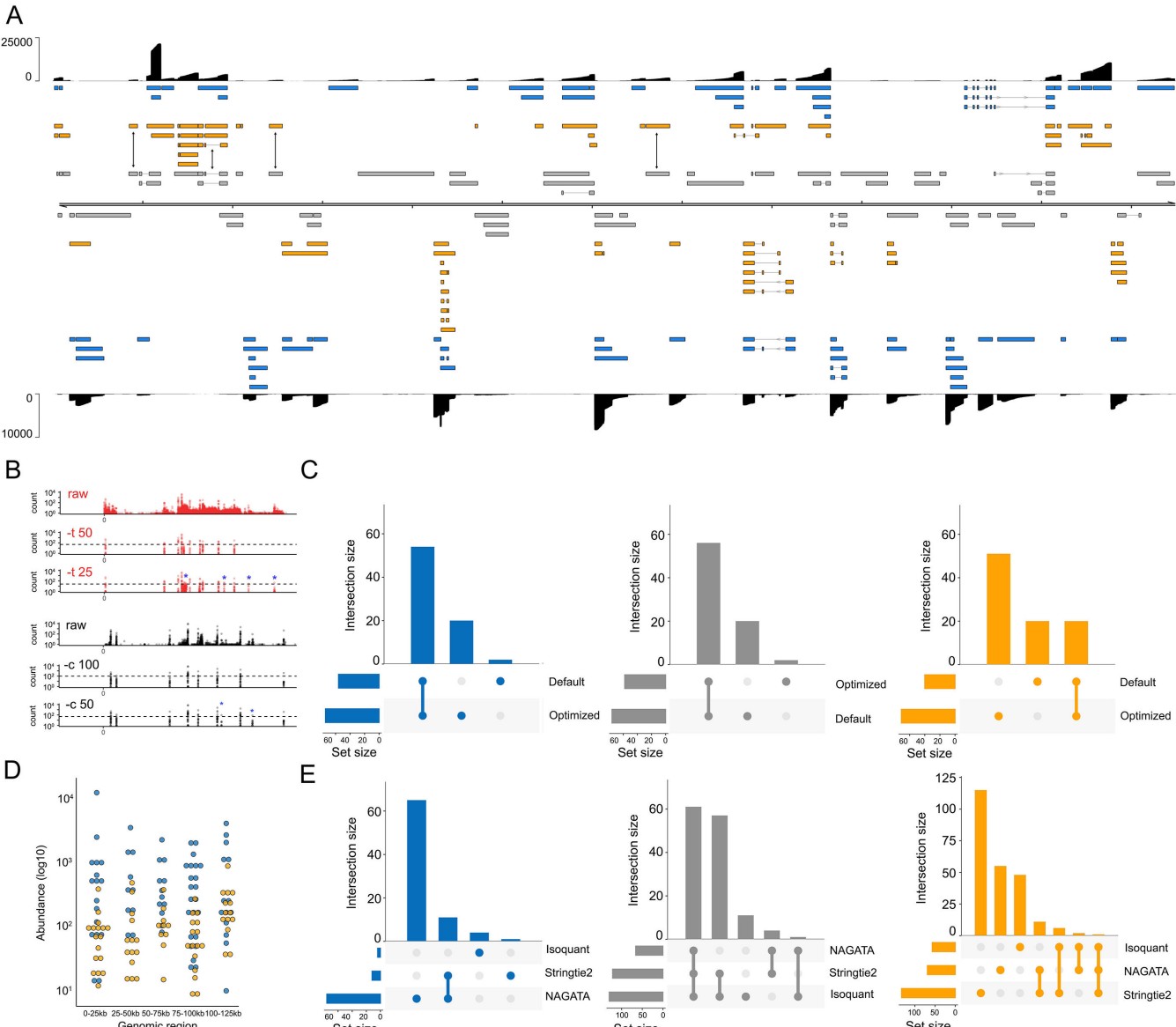

**FIG 6** Reconstructing the varicella-zoster virus transcriptome. (A) Schematic depicting the VZV strain Dumas transcriptome as constructed by NAGATA using a single previously published DRS data set (14). Read coverage is shown for each strand (black) with the y-axis denoting read depth. Transcripts are colored according to classification (orange = not recorded in existing annotation, blue = recorded in existing annotation, gray = reported in existing annotation but not detected by NAGATA in this data set). Wide and thin boxes indicate canonical CDS domains and UTRs, respectively. (B) Close-up of the forward strand TSS (red) and CPAS (black) pileups across the first 25 kb of the VZV genome. Tracks shown include the raw (pre-filtered) data and the effect of filtering using different values for TSS (-t) and -c. (C) Upset plot denoting the number of transcripts reported by NAGATA using the default and VZV-optimized configurations. (D) For each detected transcript in a given transcription unit, a raw abundance count was generated using NAGATA and color-coded according to transcript classification. For simplicity, the dot plot is divided into 25 kb windows. (E) Upset plots denoting the number of transcripts reported by NAGATA, StringTie2 (18), and Isoquant (20), segregated according to (blue) overlaps with existing annotation, (gray) not detected, and (red) not present in original annotation.

contained a 3′ junction between those of sgRNAs encoding the M and N proteins while the second used the same 3′ junction as the sgRNA encoding N protein but contained additional sequence in the 5′ leader. While Stringtie2 was also able to reconstruct all sgRNAs (and the gRNA), it also reported a larger number of artifact transcripts that did not coincide with sgRNA junctions (Fig. 7C). By contrast, Isoquant was only able to reconstruct three sgRNAs and produced over 20 novel transcripts that were not supported by the underlying data (Fig. 7C).

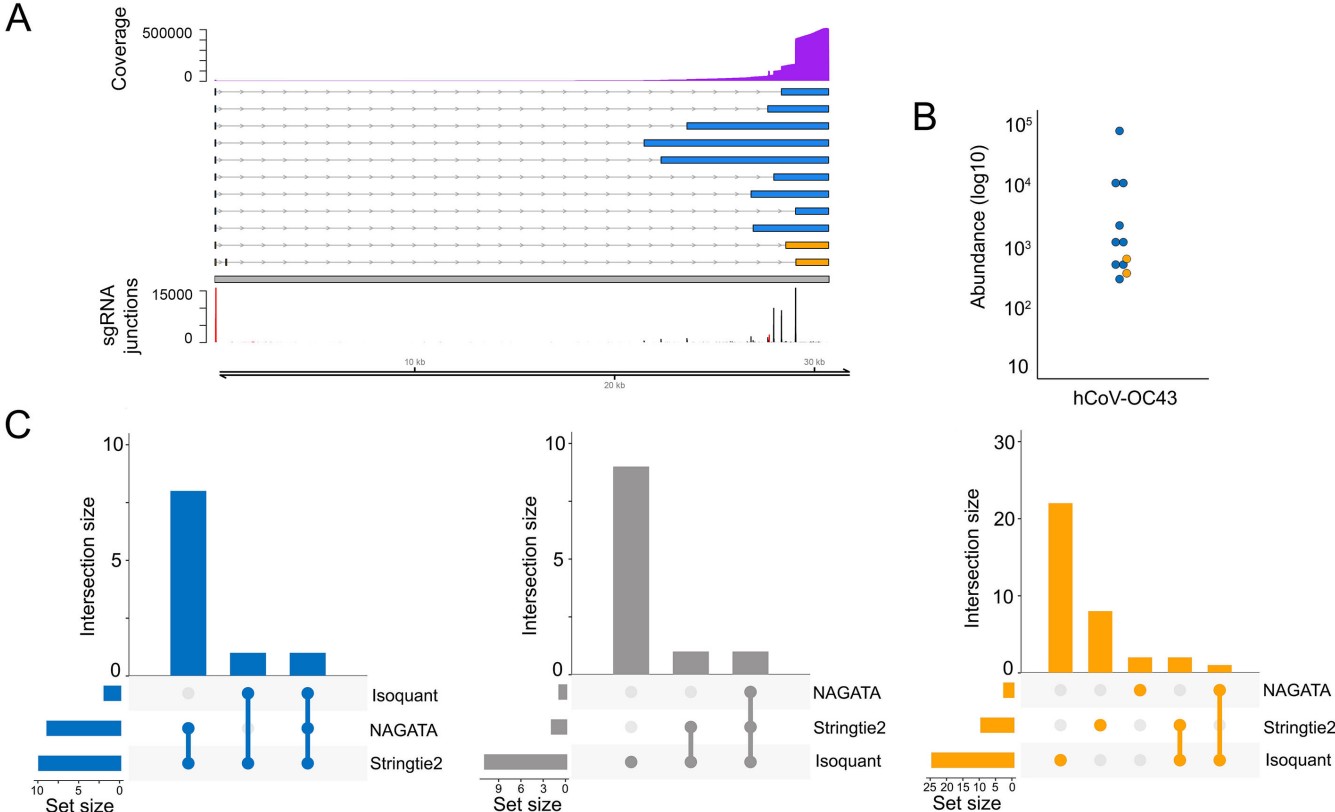

**FIG 7** Reconstructing the hCoV-OC43 transcriptome. (A) Schematic depicting the hCoV-OC43 transcriptome as constructed by NAGATA using a single previously published DRS data set (25). Read coverage is shown (purple) with the y-axis denoting read depth while the locations and abundances of sgRNA 5′ (red) and 3′ (black) junctions are also shown. Transcripts are colored according to classification (orange = not recorded in existing annotation, blue = recorded in existing annotation, gray = reported in existing annotation but not detected by NAGATA in this data set). Wide and thin boxes indicate canonical CDS domains and UTRs, respectively. (B) For each transcript, a raw abundance count was generated using NAGATA and color-coded according to transcript classification. (C) Upset plots denoting the number of transcripts reported by NAGATA, Stringtie2, and Isoquant, segregated according to (blue) overlaps with existing annotation, (gray) not detected, and (red) not present in original annotation.

## Defining the transcriptome of human adenovirus F serotype 41

The linear dsDNA genome of HAdV-F41 has a length of 34,188 bp and prior studies have indicated it shares a similar overall transcriptome architecture to other human adenoviruses (48), although many proteins are poorly conserved and differ in length (49–53). Despite increasing interest in this neglected human pathogen, the existing reference genome annotation (ON561778.1) contains just 33 computationally predicted coding sequences (CDS). To address this, we infected A549 cells with HAdV-F41 at an MOI of 50 and collected total RNA at 12, 24, and 48 hpi. We isolated the poly(A) fractions and prepared multiplexed nanopore DRS libraries using the deeplexicon protocol (30) and sequenced these for 24 h on a nanopore MinION. Following basecalling and demultiplexing, the data sets were aligned against the HAdV-F41 reference genome and processed using NAGATA. We observed 10-fold fewer read alignments against the reverse strand compared to the forward strand (Fig. 8B) and thus analyzed each strand individually with different values for -t and -c. In total, NAGATA reported 11 transcription units comprising a total of 77 transcripts, 70 on the forward strand and 7 on the reverse strand (Fig. 8A). On the forward strand, transcripts representing all major transcription units (E1A-B, L1-L5, E3) were identified and the only computationally annotated CDS that could not be assigned to NAGATA-derived transcripts were E3-14.5K and E3-14.7K. Visual inspection of read alignments in this region identified no valid transcripts that might contain these CDS, most likely indicating a need for increased sequencing depths.

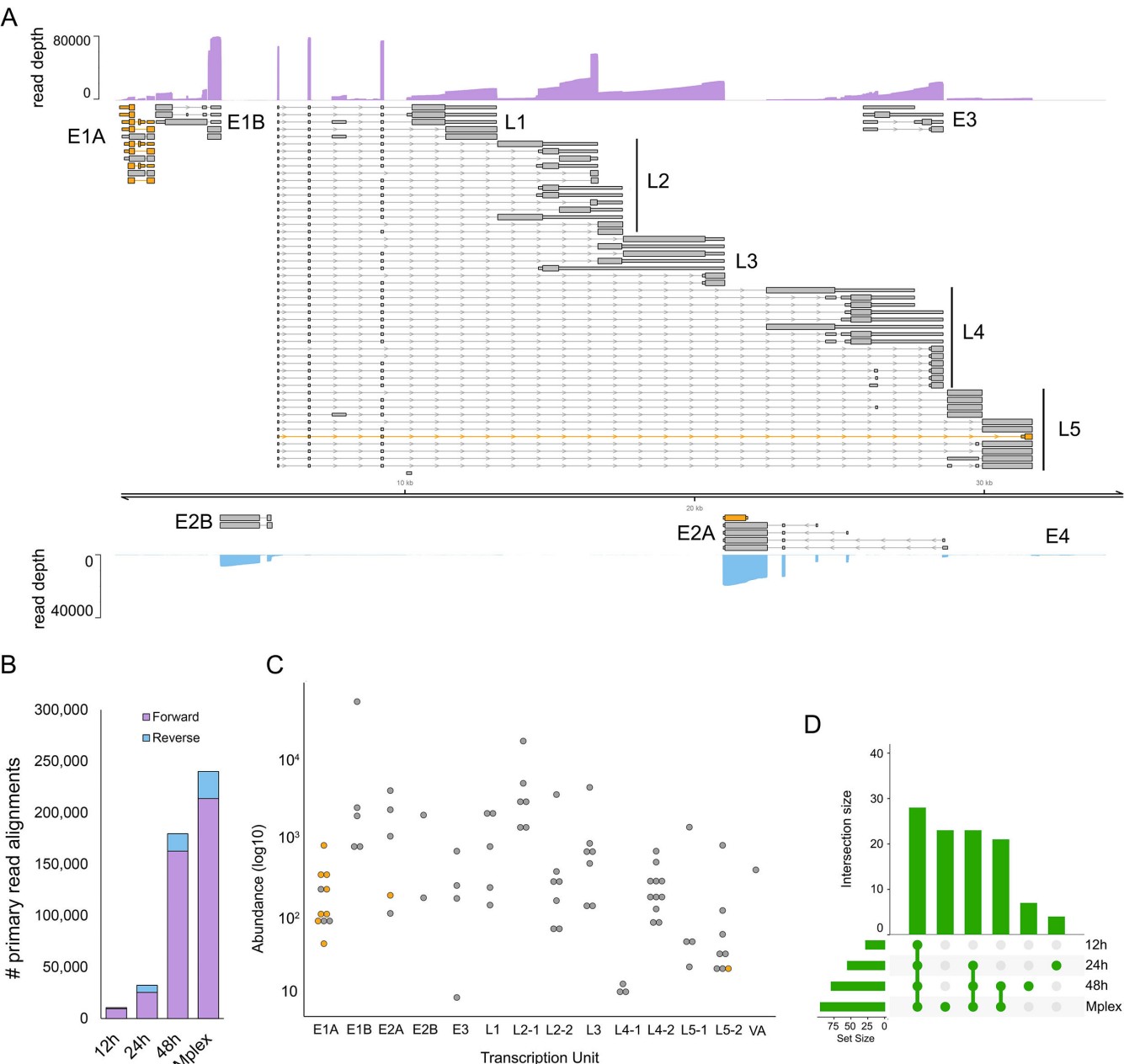

**FIG 8** Annotation of the HAdV-F41 transcriptome. (A) The reannotated HAdV-F41 transcriptome encodes 77 transcripts, of which 67 encodes 23 canonical ORFs with their UTRs defined (gray). Wide and thin boxes indicate canonical CDS domains and UTRs, respectively. A further nine transcripts (orange), putatively encoded novel or N' terminal truncated protein isoforms. Nanopore DRS coverage plots (purple) are shown for the combined (Mplex) data set (12 + 24 + 48 hpi) in a strand-specific manner. Y-axis values indicate the read depth. (B) The number of HAdV-F41 read alignments recorded against the forward and reverse strands, separated by data set, show a strong bias toward transcription from the forward strand. (C) For each detected transcript in each transcription unit, a raw abundance count was generated using NAGATA and color-coded according to transcript classification. (D) Upset plots denoting the number of transcripts reported by NAGATA in each of the individuals, as well as the merged, data sets.

We assigned CDS sequences for E1A-S and E1A-9s homologs that were not reported in the original annotation and also identified a putative N-terminal truncated Fiber$^{long}$ isoform (Fig. 8A). While the overall architecture of the HAdV-F41 transcriptome mirrors that of the HAdV-C5 transcriptome in terms of transcription units, there are some notable differences. Specifically, the L2, L4, and L5 regions all possess dual CPAS. For L2, the upstream CPAS is strong, as evidenced by the higher abundance of transcripts

terminating at this position compared to the downstream CPAS (Fig. 8C; Fig. S7). For L4 the situation is reversed with the upstream CPAS being a weak terminator (Fig. 8C; Fig. S8). By contrast, both L5 CPAS show similar strengths (Fig. 8C; Fig. S9). In contrast to HAdV-C5, the L5 region of HAdV-F41 encodes two distinct Fiber isoforms (Fiber^short and Fiber^long). For the reverse strand, we identified two discrete transcripts encoding IVa2 that differed only in their TSS position (5377 vs 5413). We further identified four alternatively spliced transcripts encoding DBP, each with a unique TSS, while a fifth single-exon transcript with a TSS in the 3′ exon of DBP, putatively encoding an N′ terminal truncated DBP, was also identified (Fig. 8A). NAGATA was unable to identify E2B transcripts encoding AdPol, TP, or any of the E4 region proteins. Consistent with previous studies of these loci (54, 55), close examination of the raw read data indicated low-level transcription of these regions but at insufficient levels for NAGATA to decode. Taken together, NAGATA has significantly increased the resolution of the hAdV41 transcriptome and provides a foundation for future studies.

## DISCUSSION

Decoding transcriptome architectures using nanopore DRS is essential for accurately interrogating RNA biology at single molecule resolution. Multiple approaches have been developed for this purpose and softwares such as Bambu (19), Isoquant (20), and StringTie2 (18) are highly effective in reconstructing transcriptome annotation for gene sparse higher eukaryotic transcriptomes (20). However, as shown by our analyses here, these appear generally unsuited to reconstructing transcriptomes of gene-dense genomes (e.g., viral genomes). This manifests as a failure to correctly separate transcript isoforms that share significant overlap and we interpret this limitation as being due to the presence of many overlapping RNAs with distinct TSS and co-terminal 3′ ends, a feature of many gene-dense genomes.

The NAGATA method was informed by specific characteristics of nanopore DRS data sets and resulting genome-level alignments. It provides an alternative approach to pre-filtering data sets to remove alignment artifacts and to enable transcriptome annotation by grouping similar structural elements, alignment-by-alignment. Alignments with similar TSS and CPAS values are used to define the initial set of transcriptional units with the most abundant TSS, and CPAS positions are used as anchors to collapse and correct all relevant alignments. The collapsing and correction steps increase sensitivity without biasing the identification of TSS and CPAS, as evidenced by comparisons to CAGE-seq and ContextMap2 results from prior studies (44, 56, 57). Similarly, isoform-level deconvolution takes place by grouping alignments in a TU by the similarity of the positions and sizes of the exons present. By identifying and retaining only reads from full-length RNAs, between 40% and 60% of reads in most data sets are removed prior to analysis. This naturally limits the utility of NAGATA in settings where read depth is low (e.g., the E4 region of HAdV-F41, Fig. 8) unless integrated with approaches such as Nanopore ReCappable Sequencing (58), that preferentially capture full-length RNAs. Similarly, it is worth reiterating that the current inability of DRS to capture the terminal 5–10 nt where 5′ caps (e.g., m7G) are present precludes the precise identification of TSS but rather produces a proximal annotation (43). In most situations, such an annotation will suffice but for studies of transcription initiation or epitranscriptomic profiling (of the first 5–10 nt), it remains necessary to adopt alternative (e.g., Nanopore ReCappable Sequencing) and/or orthologous approaches (e.g., CAGE-Seq) (58, 59). While NAGATA retains many adjustable parameters, we generally recommend only changing the TSS (-t) and CPAS (-c) abundance values along with the minimum transcript abundance (-m) flags as these are specifically sensitive to sequencing depth. Tuning these parameters is relatively simple (Fig. 6B; Fig. S5) but will always result in a trade-off between sensitivity and accuracy. Thus, for rare yet potentially (biologically) relevant transcripts, the simplest solution remains to increase sequencing depth or to integrate orthologous data sets (e.g., CAGE-Seq). The remaining parameters generally function well with their default

values in all organisms tested (human, herpesvirus, adenovirus, coronavirus) but this may not be the case for other viral species.

A limitation of all annotation tools, including NAGATA, is the underlying assumption that the transcriptome architecture remains consistent across a genome. While this holds in many cases (e.g., *H. sapiens*, adenoviruses, coronaviruses), there are notable exceptions. The transcriptome architecture of VZV is dominated by single-exon transcripts with co-terminal 3′ ends. However, there are also several small regions encoding multitudes of alternatively spliced multi-exon transcripts (Fig. 6). Accurately reconstructing "transcriptional islands" such as this may require different parameters to the rest of the genome. Similarly, combining multiple data sets (e.g., an infection time course) may increase sensitivity (Fig. 5C) although our general recommendation is to analyze all available data sets individually and in combination.

The development of NAGATA enabled us to generate a substantially improved annotation of the HAdV-F41. Here, the existing annotation comprised a handful of predicted CDS with no information on TSS or CPAS. Using NAGATA, we identified 77 transcript isoforms from 11 TUs and were able to assign new or known CDS to all of these. We further observed the presence of CPAS redundancies for the L2, L4, and L5 TUs (Fig. 8) that are not seen in the related HAdV-C5. The functional relevance of these is not known and thus bears further investigation.

In summary, NAGATA offers a novel and flexible approach to generating high-resolution transcriptome annotations from nanopore DRS data sets that can be applied against both gene-sparse and gene-dense organisms. Given increasing global efforts to sequence a large number of genomes from all domains of life and the need to supplement these with accurate transcriptome maps, we offer NAGATA as a new approach to achieve this objective.

## ACKNOWLEDGMENTS

S.S. was funded by the Deutsche Forschungsgemeinschaft (DFG, German Research Foundation) in the framework of the Research Unit FOR5200 DEEP-DV (443644894) project 08. A.C.W. is supported by grants from the National Institute of Allergy and Infectious Disease R01-AI170583 and R01AI176335. D.P.D. is supported by a German Centre for Infection Research (DZIF) Associate Professorship and the NIAID grants R01-AI170583 and R01-AI152543. D.P.D. and S.S. also receive funding from the Deutsche Forschungsgemeinschaft (DFG, German Research Foundation) under Germany's Excellence Strategy—EXC 2155—project number 390874280.

## AUTHOR AFFILIATIONS

[1]Department of Microbiology, New York University School of Medicine, New York, New York, USA

[2]Institute of Virology, Hannover Medical School, Hannover, Germany

[3]Institute of Virology, University Medical Center, Albert-Ludwigs-University Freiburg, Freiburg, Germany

[4]German Center for Infection Research (DZIF), partner site Hannover-Braunschweig, Hannover, Germany

[5]Cluster of Excellence RESIST (EXC 2155), Hannover Medical School, Hannover, Germany

## AUTHOR ORCIDs

Jonathan S. Abebe http://orcid.org/0009-0000-1268-0427
Yasmine Alwie http://orcid.org/0000-0002-7001-5641
Erik Fuhrmann http://orcid.org/0009-0001-8512-9369
Julia Mai http://orcid.org/0000-0001-9501-5691
Ruth Verstraten http://orcid.org/0009-0008-6046-6335

Sabrina Schreiner ⓘ http://orcid.org/0000-0002-5744-7159
Angus C. Wilson ⓘ http://orcid.org/0000-0002-5016-4164
Daniel P. Depledge ⓘ http://orcid.org/0000-0002-4292-0599

## FUNDING

| Funder | Grant(s) | Author(s) |
|---|---|---|
| HHS \| NIH \| National Institute of Allergy and Infectious Diseases (NIAID) | R01-AI170583 | Angus C. Wilson |
| | | Daniel P. Depledge |
| HHS \| NIH \| National Institute of Allergy and Infectious Diseases (NIAID) | R01-AI170583 | Angus C. Wilson |
| HHS \| NIH \| National Institute of Allergy and Infectious Diseases (NIAID) | R01-AI152543 | Daniel P. Depledge |
| Deutsche Forschungsgemeinschaft (DFG) | FOR5200 DEEP-DV (443644894) | Sabrina Schreiner |
| Deutsche Forschungsgemeinschaft (DFG) | EXC 2155 (390874280) | Sabrina Schreiner |
| | | Daniel P. Depledge |

## DATA AVAILABILITY

NAGATA is written in Python 3 and is available in the https://github.com/DepledgeLab/NAGATA repository, along with test datasets and accessory scripts. The deeplexicon multiplexed raw Fast5 dataset generated for HAdV-F41 is available from the ENA/SRA under the accession number PRJEB72818. BAM files used as inputs for the annotation tools used here are available via FigShare along with relevant outputs under the accessions: https://doi.org/10.6084/m9.figshare.25897417.v1, https://doi.org/10.6084/m9.figshare.25897702.v1, https://doi.org/10.6084/m9.figshare.25897453.v1, https://doi.org/10.6084/m9.figshare.25897534.v1, and https://doi.org/10.6084/m9.figshare.25897681.v1.

## ADDITIONAL FILES

The following material is available online.

### Supplemental Material

**Figure S1 (mSystems00505-24-S0001.tif).** Further characteristics of nanopore DRS alignments.
**Figure S2 (mSystems00505-24-S0002.tif).** Adenovirus type 5 transcriptome reconstructions from synthetic data sets.
**Figure S3 (mSystems00505-24-S0003.tif).** Varicella zoster virus transcriptome reconstructions from a synthetic data set.
**Figure S4 (mSystems00505-24-S0004.tif).** Adenovirus type 5 transcriptome reconstructions using a real DRS data set.
**Figure S5 (mSystems00505-24-S0005.tif).** Analysis of TSS/CPAS identification using default and VZV-optimized NAGATA settings.
**Figure S6 (mSystems00505-24-S0006.tif).** Varicella zoster virus transcriptome reconstructions using a real DRS data set.
**Figure S7 (mSystems00505-24-S0007.tif).** Sequence level analysis of HAdV-F41 L2_1 and L2_1 CPAS.
**Figure S8 (mSystems00505-24-S0008.tif).** Sequence level analysis of HAdV-F41 L4_1 and L4_1 CPAS.
**Figure S9 (mSystems00505-24-S0009.tif).** Sequence level analysis of HAdV-F41 L5_1 and L5_1 CPAS.
**Table S1 (mSystems00505-24-S00010.xlsx).** Data sets used in this study.

Open Peer Review

PEER REVIEW HISTORY (review-history.pdf). An accounting of the reviewer comments and feedback.

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
