## [Reviewer comments · mSystems]

Nanopore Guided Annotation of Transcriptome Architectures

Jonathan Abebe, Yasmine Alwie, Erik Fuhrmann, Jonas Leins, Julia Mai, Ruth Verstraten, Sabrina Schreiner, Angus Wilson, and Daniel Depledge

Corresponding Author(s): Daniel Depledge, Medizinische Hochschule Hannover

Review Timeline:

Submission Date:	April 9, 2024
Editorial Decision:	May 17, 2024
Revision Received:	May 30, 2024
Accepted:	June 11, 2024

Editor: Evelien Adriaenssens

Reviewer(s): The reviewers have opted to remain anonymous.

Transaction Report:

DOI: <https://doi.org/10.1128/msystems.00505-24>

Re: mSystems00505-24 (Nanopore Guided Annotation of Transcriptome Architectures)

Dear Prof. Daniel P Depledge:

The reviewers have raised some excellent comments regarding benchmarking and the availability of data, which I would like to see addressed. I also agree with the comment from reviewer 2 that abstract is not representative of the actual content of the manuscript. It would be great to see this rewritten, and including the NAGATA name will also increase the programme's visibility.

Regarding the data availability: The authors have provided a BioProject accession number for the new sequence data generated for this manuscript, but the BioProject itself is not available. The project and all underlying sequence data should be made publicly available in the correct formats before the manuscript can be accepted for publication.

Revision Guidelines

Sincerely,
Evelien Adriaenssens
Editor
mSystems

Reviewer #1 (Comments for the Author):

The manuscript by Abebe and colleagues presents a novel computational tool, NAGATA, designed to generate high-resolution transcriptome annotations using Nanopore direct RNA sequencing (DRS) datasets. The study focuses on addressing the challenges of annotating gene dense viral genomes and demonstrates NAGATA's efficacy through comprehensive benchmarking against existing tools using synthetic and real-world datasets. The manuscript is well-written and offers significant contributions to the field of virology and transcriptomics.

The development of NAGATA is a significant advancement in transcriptome annotation, especially for gene dense viral genomes. The tool's ability to handle complex transcriptome architectures is commendable.

The manuscript includes thorough benchmarking against existing tools, such as Stringtie2, Bambu, and Isoquant, using both synthetic and real datasets. This demonstrates NAGATA's superior performance in identifying transcript isoforms in gene dense genomes.

The authors provide links to datasets and the NAGATA code repository, promoting transparency and reproducibility.

I believe that it would strengthen the study to include a comparison with additional annotation tools like LoRTIA, Talon, Squanti, and MOP2. This would provide a more comprehensive evaluation of NAGATA's performance, except if the authors can explain why the three programs they used are sufficient for the evaluation.

The accuracy of TSS and CPAS identification, given the 5-10 nt discrepancy at the 5' end, could be elaborated upon. Discussing the implications of this discrepancy and how it impacts downstream analyses would be beneficial.

Discussing the potential use of other poly(A) tail length determination tools such as tailfinder, and comparing their performance with Nanopolish, could add value to the study.

It should be discussed how NAGATA ensures that low-abundance, yet potentially biologically significant, transcripts are not filtered out during the pre-filtering and final filtering steps. Could there be a mechanism to flag these for further investigation?

- Reference Annotation and Validation: In the absence of reference annotations and supplementary methods, how can we be sure that the identified TSSs are accurate, given the missing 5-15 nt at the 5' end of the DRS reads?
- Cause of Missing 5-10 nt: Why is there typically a 5-10 nt missing from the 5' end of DRS reads? How is it possible that transcripts of very different lengths consistently miss approximately the same portion at their 5' end?
- Virus Specificity: Is this issue of missing nucleotides at the 5' end particularly prevalent in viruses, or is it a general observation across all DRS sequencing?
- Adjusting Window Sizes for Other Organisms: For other organisms, do the authors recommend or find it necessary to modify the window sizes considering the potentially varying degrees of missing 5' ends?

It is not clear, whether the specific settings or strategies do the authors recommend for combining datasets from multiple time points or samples to avoid losing transcripts that are only present at certain times or conditions.

While the authors have tested NAGATA on human chromosome 1, how well does it perform on gene-dense regions within eukaryotic genomes? Are there plans to extend the evaluation to such regions, and if so, what preliminary results can be shared?

Comments on Data and Methods

Script Discrepancy: The article mentions the script `post_intersect_processing_v4.1.py`, but on the GitHub repository, there is only `post_intersect_processing_v3_alt.py`. Clarification is needed on whether these scripts are the same or if the repository should be updated to include the correct version.

Reproducible Comparison Method: Please provide a reproducible method to compare the NAGATA results (or any other set of transcripts) with a reference annotation. Currently, it is challenging to reproduce the results without a clear methodology.

Uploading Benchmark Data: It would be helpful if the results of the other tools and the BAM files used to generate them were uploaded as well. If the files are large, you can use platforms like Figshare to share these datasets. This will facilitate the reproducibility and verification of your results.

Biological Relevance of Identified Transcripts: While the technical performance of NAGATA is well-demonstrated, including some biological insights or hypotheses generated from the identified transcripts would strengthen the study's relevance.

Overall, the article presents a well-constructed and innovative approach to transcriptome annotation for gene dense viral genomes. With a few additional comparisons and clarifications, this study could significantly advance our understanding and capability in viral transcriptomics.

Reviewer #2 (Comments for the Author):

The manuscript by Abebe and colleagues deals with developing and validating the new computational approach, NAGATA, to

improve the DRS analysis. The topic is important considering the enormous data flow from the DRS, which often generates "unrealistic" and "weird" data sets. Hence, all efforts to improve the computational approaches are welcome as they make the data sets more useful and understandable.

The manuscript is well-written and although I am not a bioinformatician, I can understand most of it. Very well done in this regard! And for sure the NAGATA will help to understand the beauty and pain of the DRS.

However, I do have several comments, which I hope the authors will consider to improve the present manuscript text.

*) The abstract is a bit misleading as it gives the impression that HAdV-F41 is the main part of it, although it is not. So, the authors should rephrase it and include a statement that data analysis was also done on HAdV-C5, VZV, and HCoV-OC43. Why not even to mention the NAGATA in the abstract, aren't you proud of it?

*)page 7: It remains unclear to me why the authors have decided to use HAdV-C5 dataset from the Weitzman's lab. There are at least 2 other Nanopore data sets from HAdV-C5/-C2 (Donovan-Banfield., 2020 and Jakobsson., 2021) available. It would be nice understand the rationale behind it.

*) page 3: I recommend the authors include the review by Grand (doi: 10.1080/21505594.2023.2242544) in the section where they describe HAdV-F41. This is absolutely best review (and very recent!) about HAdV-F41, and it will not hurt even the authors to read it.

*)The authors should check the spelling throughout the manuscript. Is it HAdV-F41 (page 12) or hAdV41 (Fig. 8)? The same applies to hours post infection vs. hpi. If the acronym is defined at the beginning, there is no need to go back to long terms again.

*)Page 13: I would like the authors to speculate why "identified and the only annotated CDS that could not be assigned to NAGATA-derived transcripts were E3-14.5K and E3-14.7K". Is it a technical issue?

*)Page 13: the CPAS redundancy is a nice finding. I suggest the authors make a supplementary illustration of the identified CPAS at the L2 at the sequence level. That would be very informative when studying the details of the L2 alternative polyadenylation.

*Page 13: "NAGATA was unable to identify transcripts encoding AdPol, TP,". This is actually known from the Jakobsson., 2021 study that E2B transcripts are, in general, difficult to detect. If one is even more correct, the original finding about it comes from the study by Stillman et al., 40 years ago (DOI: 10.1016/0092-8674(81)90145-8.). Consider including these references to tell the reader that similarly to HAdV-C2, in HAdV-F41 the E2B transcripts are very rare.

*)Fig.8: The authors say: "further 9 transcripts (pink)", however, I can not see that these 9 transcripts are pink, rather dark yellow, or am I looking at wrong transcripts?

*)Discussion page 14: "CPAS redundancies for the L2, L4, and L5 TUs (Fig. 7)...", I guess that the authors mean Fig 8?

The manuscript by Abebe and colleagues presents a novel computational tool, NAGATA, designed to generate high-resolution transcriptome annotations using Nanopore direct RNA sequencing (DRS) datasets. The study focuses on addressing the challenges of annotating gene dense viral genomes and demonstrates NAGATA's efficacy through comprehensive benchmarking against existing tools using synthetic and real-world datasets. The manuscript is well-written and offers significant contributions to the field of virology and transcriptomics.

The development of NAGATA is a significant advancement in transcriptome annotation, especially for gene dense viral genomes. The tool's ability to handle complex transcriptome architectures is commendable.

The manuscript includes thorough benchmarking against existing tools, such as Stringtie2, Bambu, and Isoquant, using both synthetic and real datasets. This demonstrates NAGATA's superior performance in identifying transcript isoforms in gene dense genomes.

The authors provide links to datasets and the NAGATA code repository, promoting transparency and reproducibility.

I believe that it would strengthen the study to include a comparison with additional annotation tools like LoRTIA, Talon, Squanti, and MOP2. This would provide a more comprehensive evaluation of NAGATA's performance, except if the authors can explain why the three programs they used are sufficient for the evaluation.

The accuracy of TSS and CPAS identification, given the 5-10 nt discrepancy at the 5' end, could be elaborated upon. Discussing the implications of this discrepancy and how it impacts downstream analyses would be beneficial.

Discussing the potential use of other poly(A) tail length determination tools such as tailfinder, and comparing their performance with Nanopolish, could add value to the study.

It should be discussed how NAGATA ensures that low-abundance, yet potentially biologically significant, transcripts are not filtered out during the pre-filtering and final filtering steps. Could there be a mechanism to flag these for further investigation?

- Reference Annotation and Validation: In the absence of reference annotations and supplementary methods, how can we be sure that the identified TSSs are accurate, given the missing 5-15 nt at the 5' end of the DRS reads?
- Cause of Missing 5-10 nt: Why is there typically a 5-10 nt missing from the 5' end of DRS reads? How is it possible that transcripts of very different lengths consistently miss approximately the same portion at their 5' end?
- Virus Specificity: Is this issue of missing nucleotides at the 5' end particularly prevalent in viruses, or is it a general observation across all DRS sequencing?
- Adjusting Window Sizes for Other Organisms: For other organisms, do the authors recommend or find it necessary to modify the window sizes considering the potentially varying degrees of missing 5' ends?

It is not clear, whether the specific settings or strategies do the authors recommend for combining datasets from multiple time points or samples to avoid losing transcripts that are only present at certain times or conditions.

While the authors have tested NAGATA on human chromosome 1, how well does it perform on gene-dense regions within eukaryotic genomes? Are there plans to extend the evaluation to such regions, and if so, what preliminary results can be shared?

Comments on Data and Methods

Script Discrepancy: The article mentions the script `post_intersect_processing_v4.1.py`, but on the GitHub repository, there is only `post_intersect_processing_v3_alt.py`. Clarification is needed on whether these scripts are the same or if the repository should be updated to include the correct version.

Reproducible Comparison Method: Please provide a reproducible method to compare the NAGATA results (or any other set of transcripts) with a reference annotation. Currently, it is challenging to reproduce the results without a clear methodology.

Uploading Benchmark Data: It would be helpful if the results of the other tools and the BAM files used to generate them were uploaded as well. If the files are large, you can use platforms like Figshare to share these datasets. This will facilitate the reproducibility and verification of your results.

Biological Relevance of Identified Transcripts: While the technical performance of NAGATA is well-demonstrated, including some biological insights or hypotheses generated from the identified transcripts would strengthen the study's relevance.

Overall, the article presents a well-constructed and innovative approach to transcriptome annotation for gene dense viral genomes. With a few additional comparisons and clarifications, this study could significantly advance our understanding and capability in viral transcriptomics.

Comments from the Editor:

The reviewers have raised some excellent comments regarding benchmarking and the availability of data, which I would like to see addressed. I also agree with the comment from reviewer 2 that abstract is not representative of the actual content of the manuscript. It would be great to see this rewritten, and including the NAGATA name will also increase the programme's visibility.

We appreciate the comments from the editor and reviewers and have sought to address all of these below, including a complete rewrite of the abstract to better reflect the content and major points of the paper.

Regarding the data availability: The authors have provided a BioProject accession number for the new sequence data generated for this manuscript, but the BioProject itself is not available. The project and all underlying sequence data should be made publicly available in the correct formats before the manuscript can be accepted for publication.

We apologize for the oversight in not making this available at the time of submission. The BioProject and sequence data are now publicly available.

Reviewer #1 (Comments for the Author):

The manuscript by Abebe and colleagues presents a novel computational tool, NAGATA, designed to generate high-resolution transcriptome annotations using Nanopore direct RNA sequencing (DRS) datasets. The study focuses on addressing the challenges of annotating gene dense viral genomes and demonstrates NAGATA's efficacy through comprehensive benchmarking against existing tools using synthetic and real-world datasets. The manuscript is well-written and offers significant contributions to the field of virology and transcriptomics.

The development of NAGATA is a significant advancement in transcriptome annotation, especially for gene dense viral genomes. The tool's ability to handle complex transcriptome architectures is commendable.

The manuscript includes thorough benchmarking against existing tools, such as Stringtie2, Bambu, and Isoquant, using both synthetic and real datasets. This demonstrates NAGATA's superior performance in identifying transcript isoforms in gene dense genomes.

The authors provide links to datasets and the NAGATA code repository, promoting transparency and reproducibility.

We appreciate the reviewer's enthusiasm for our work and the constructive critiques provided below. We genuinely feel that these comments have enabled us to further improve our manuscript.

I believe that it would strengthen the study to include a comparison with additional annotation tools like LoRTIA, Talon, Squanti, and MOP2. This would provide a more comprehensive evaluation of NAGATA's performance, except if the authors can explain why the three programs they used are sufficient for the evaluation.

We discussed benchmarking extensively during the development of NAGATA and decided to only include transcriptome annotation softwares that had undergone peer review. At the time of writing, LoRTIA has no publication associated with it while Talon has remained a preprint since 2020. Master of Pores 2 is not designed for transcriptome reconstruction. Squanti and Squanti2 were recently integrated, together with new features, into Squanti3 and offers a powerful approach for curation of transcriptome annotations derived from long-read sequencing datasets. While potentially useful for evaluating NAGATA outputs on higher eukaryotic transcriptomes, the underlying models do not appear appropriate for gene dense organisms. Bambu, Isoquant, and StringTie2 are all popular peer-reviewed transcriptome annotation softwares that have been demonstrated to significantly outperform other approaches and thus remain the best possible comparators for NAGATA.

The accuracy of TSS and CPAS identification, given the 5-10 nt discrepancy at the 5' end, could be elaborated upon. Discussing the implications of this discrepancy and how it impacts downstream analyses would be beneficial.

This is an excellent point and we have now addressed this in both the results section (page 7) and discussion (page 15).

Discussing the potential use of other poly(A) tail length determination tools such as tailfinder, and comparing their performance with Nanopolish, could add value to the study.

At the time of writing the code for NAGATA, tailfindR and nanopolish were the only effective softwares for classifying reads according to the presence of readable poly(A) tails and both showed a high correlation in their results (PMID: 31266821). Nanopolish was chosen as this is simpler to integrate into a wider workflow. Looking to the future, this information can now also be obtained directly via the dorado basecaller provided by nanopore although this is still under active development and producing highly inconsistent results between releases. Once this situation stabilises then we will certainly explore integrating this into NAGATA.

It should be discussed how NAGATA ensures that low-abundance, yet potentially biologically significant, transcripts are not filtered out during the pre-filtering and final filtering steps. Could there be a mechanism to flag these for further investigation?

Any aligned DRS dataset will contain a certain level of artefact derived from misaligned and incomplete (i.e. 5' degraded) reads. NAGATA provides numerous flags to assist in

reducing this noise which can lead to the identification of potentially biologically relevant low abundance transcripts. However, optimal results are best secured by increasing sequencing depth and integration orthologous datasets. This is now discussed on Page 15.

- Reference Annotation and Validation: In the absence of reference annotations and supplementary methods, how can we be sure that the identified TSSs are accurate, given the missing 5-15 nt at the 5' end of the DRS reads?

This is a good point and it is important to be clear that we are not claiming to identify the precise TSS but rather the proximal location of the TSS. We have made this clearer in the discussion (page 15)

- Cause of Missing 5-10 nt: Why is there typically a 5-10 nt missing from the 5' end of DRS reads? How is it possible that transcripts of very different lengths consistently miss approximately the same portion at their 5' end?

Briefly, the presence of 5' caps prevents the nanopore from sequencing the terminal 5-10 nt (PMID: 34428294). We have clarified this in the first part of the results section (page 7)

- Virus Specificity: Is this issue of missing nucleotides at the 5' end particularly prevalent in viruses, or is it a general observation across all DRS sequencing?

This is indeed a general issue that impacts all capped RNAs. We have clarified this in the first part of the results section (page 7)

Adjusting Window Sizes for Other Organisms: For other organisms, do the authors recommend or find it necessary to modify the window sizes considering the potentially varying degrees of missing 5' ends?

In general we recommend only adjusting the flags that are impacted by sequencing depth (-t, -c, -m). The default values associated with all remaining flags seem to perform well across multiple species (human, adenovirus, coronavirus, herpesvirus) but this may not always be the case for the other viruses. We have detailed this in the discussion (page 15).

It is not clear, whether the specific settings or strategies do the authors recommend for combining datasets from multiple time points or samples to avoid losing transcripts that are only present at certain times or conditions.

Our recommendation is to apply NAGATA to both individual and combined datasets, as demonstrated in Figure 5. We have made this clearer in the discussion (page 15).

While the authors have tested NAGATA on human chromosome 1, how well does it perform on gene-dense regions within eukaryotic genomes? Are there plans to extend the evaluation to such regions, and if so, what preliminary results can be shared?

We have not explicitly tested NAGATA on (comparatively) gene dense regions of eukaryotic genomes but would note that this gene-density is a somewhat subjective term. For instance, the p15.5 region on human chromosome 11 is considered gene dense (PMID: 14656967) with around 40 genes / Mb but even in this context, genes exist as islands, remaining segregated with large intergenic region (> 2kb) between them, and thus would be very different to viruses in which genes overlap significantly. As such, the expected NAGATA outputs for the p15.5 would not be any different versus (comparatively) gene sparse regions in the same genome.

Comments on Data and Methods

Script Discrepancy: The article mentions the script `post_intersect_processing_v4.1.py`, but on the GitHub repository, there is only `post_intersect_processing_v3_alt.py`. Clarification is needed on whether these scripts are the same or if the repository should be updated to include the correct version.

We appreciate the reviewer bringing this to our attention and have uploaded the correct version of this script to the GitHub page.

Reproducible Comparison Method: Please provide a reproducible method to compare the NAGATA results (or any other set of transcripts) with a reference annotation. Currently, it is challenging to reproduce the results without a clear methodology.

We have updated the NAGATA GitHub page to include an example dataset and accompanying workflow so that users can verify that NAGATA is working correctly. Further, we have uploaded the BAM files used in this study to FigShare (see below) along with relevant outputs to ensure that the analyses presented here can be reproduced.

Uploading Benchmark Data: It would be helpful if the results of the other tools and the BAM files used to generate them were uploaded as well. If the files are large, you can use platforms like Figshare to share these datasets. This will facilitate the reproducibility and verification of your results.

As indicated above, we have now created a FigShare accession and have made all relevant data available here with the following accessions:

Figure 4 - <https://doi.org/10.6084/m9.figshare.25897417.v1>

Figure 5 – <https://doi.org/10.6084/m9.figshare.25897702.v1>

Figure 6 – <https://doi.org/10.6084/m9.figshare.25897453.v1>

Figure 7 – <https://doi.org/10.6084/m9.figshare.25897534.v1>

Figure 8 – <https://doi.org/10.6084/m9.figshare.25897681.v1>

The data availability statement in the manuscript has been updated to reflect this.

Biological Relevance of Identified Transcripts: While the technical performance of NAGATA is well-demonstrated, including some biological insights or hypotheses generated from the identified transcripts would strengthen the study's relevance.

The reviewer is correct that obtaining additional biological insight from this work would further increase the relevance of this study. This is now an active pursuit of the lab but the reality is that hAdV41 remains difficult to work with and lacks many reagents (e.g. antibodies) that are required for deeper analyses. We would thus consider such work beyond the scope of this specific publication.

Overall, the article presents a well-constructed and innovative approach to transcriptome annotation for gene dense viral genomes. With a few additional comparisons and clarifications, this study could significantly advance our understanding and capability in viral transcriptomics.

We appreciate the reviewer's thoughtful comments.

Reviewer #2 (Comments for the Author):

The manuscript by Abebe and colleagues deals with developing and validating the new computational approach, NAGATA, to improve the DRS analysis. The topic is important considering the enormous data flow from the DRS, which often generates "unrealistic" and "weird" data sets. Hence, all efforts to improve the computational approaches are welcome as they make the data sets more useful and understandable.

The manuscript is well-written and although I am not a bioinformatician, I can understand most of it. Very well done in this regard! And for sure the NAGATA will help to understand the beauty and pain of the DRS.

However, I do have several comments, which I hope the authors will consider to improve the present manuscript text.

As with the comments from reviewer #1, we are appreciative of the reviewer's enthusiasm for our work and the constructive critiques provided.

*) The abstract is a bit misleading as it gives the impression that HAdV-F41 is the main

part of it, although it is not. So, the authors should rephrase it and include a statement that data analysis was also done on HAdV-C5, VZV, and HCoV-OC43. Why not even to mention the NAGATA in the abstract, aren't you proud of it?

The reviewer is absolutely correct and we have completely rewritten the abstract to address these points.

*)page 7: It remains unclear to me why the authors have decided to use HAdV-C5 dataset from the Weitzman's lab. There are at least 2 other Nanopore data sets from HAdV-C5/-C2 (Donovan-Banfield., 2020 and Jakobsson., 2021) available. It would be nice understand the rationale behind it.

This dataset was generated during our previous collaboration with the Weitzman lab and resulted in a manually generated high-resolution HAdV-C5 annotation that served as an excellent benchmark for testing NAGATA.

*) page 3: I recommend the authors include the review by Grand (doi: 10.1080/21505594.2023.2242544) in the section where they describe HAdV-F41. This is absolutely best review (and very recent!) about HAdV-F41, and it will not hurt even the authors to read it.

This is indeed an excellent review and we have now included this citation.

*)The authors should check the spelling throughout the manuscript. Is it HAdV-F41 (page 12) or hAdV41 (Fig. 8)? The same applies to hours post infection vs. hpi. If the acronym is defined at the beginning, there is no need to go back to long terms again.

We appreciate the reviewer spotting these discrepancies. These have been fixed.

*)Page 13: I would like the authors to speculate why" identified and the only annotated CDS that could not be assigned to NAGATA-derived transcripts were E3-14.5K and E3-14.7K". Is it a technical issue?

The challenge of working with HAdV-F41 remains the fact that viral mRNAs only account for a small proportions of all mRNAs present in the infected cell (unlike for HAdV-C5 and the other viral species used in this study). The reason we could not assign the CDS to any given transcript is most likely due to the low depth of sequencing obtained. This has been clarified in the text (results page 13)

*)Page 13: the CPAS redundancy is a nice finding. I suggest the authors make a supplementary illustration of the identified CPAS at the L2 at the sequence level. That

would be very informative when studying the details of the L2 alternative polyadenylation.

An excellent idea. We have include new supplementary figures (S7 – S9) providing this information for the L2, L4, and L5 CPAS redundancies

*Page 13: "NAGATA was unable to identify transcripts encoding AdPol, TP,". This is actually known from the Jakobsson., 2021 study that E2B transcripts are, in general, difficult to detect. If one is even more correct, the original finding about it comes from the study by Stillman et al., 40 years ago (DOI: 10.1016/0092-8674(81)90145-8.). Consider including these references to tell the reader that similarly to HAdV-C2, in HAdV-F41 the E2B transcripts are very rare.

The reviewer is absolutely correct and we apologize for not including mention of these papers previously. This has now been corrected (Page 14)

*)Fig.8: The authors say: "further 9 transcripts (pink)", however, I can not see that these 9 transcripts are pink, rather dark yellow, or am I looking at wrong transcripts?

We have corrected this error in the legend of Figure 8

*)Discussion page 14: "CPAS redundancies for the L2, L4, and L5 TUs (Fig. 7)...", I guess that the authors mean Fig 8?

Fig. 8 is indeed correct and this has now been fixed.

Re: mSystems00505-24R1 (Nanopore Guided Annotation of Transcriptome Architectures)

Dear Prof. Daniel P Depledge:

Thank you for addressing the reviewer's comments and mine. There is one minor issue remaining, but I do not want to hold up the publication of the article.

Please check the data availability in BioProject PRJEB72818. While the project repository is publicly available, the content is empty. This should be addressed immediately.

Your manuscript has been accepted, and I am forwarding it to the ASM production staff for publication. Your paper will first be checked to make sure all elements meet the technical requirements. ASM staff will contact you if anything needs to be revised before copyediting and production can begin. Otherwise, you will be notified when your proofs are ready to be viewed.

Sincerely,
Evelien Adriaenssens
Editor
mSystems